# Evaluating the Metabolomic Profile and Anti-Pathogenic Properties of *Cannabis* Species

**DOI:** 10.3390/metabo14050253

**Published:** 2024-04-26

**Authors:** Shadrack Monyela, Prudence Ngalula Kayoka, Wonder Ngezimana, Lufuno Ethel Nemadodzi

**Affiliations:** 1Department of Agriculture and Animal Health, University of South Africa, Science Campus, Florida, Johannesburg 1710, South Africa; 2Department of Horticulture, Faculty of Plant and Animal Sciences and Technology, Marondera University of Agricultural Sciences and Technology, Marondera P.O. Box 35, Zimbabwe

**Keywords:** *Cannabis*, metabolomic profile, phytocannabinoids, antimicrobial properties, antibacterial properties, phytopathogens, fish pathogens

## Abstract

The *Cannabis* species is one of the potent ancient medicinal plants acclaimed for its medicinal properties and recreational purposes. The plant parts are used and exploited all over the world for several agricultural and industrial applications. For many years *Cannabis* spp. has proven to present a highly diverse metabolomic profile with a pool of bioactive metabolites used for numerous pharmacological purposes ranging from anti-inflammatory to antimicrobial. *Cannabis sativa* has since been an extensive subject of investigation, monopolizing the research. Hence, there are fewer studies with a comprehensive understanding of the composition of bioactive metabolites grown in different environmental conditions, especially *C. indica* and a few other Cannabis strains. These pharmacological properties are mostly attributed to a few phytocannabinoids and some phytochemicals such as terpenoids or essential oils which have been tested for antimicrobial properties. Many other discovered compounds are yet to be tested for antimicrobial properties. These phytochemicals have a series of useful properties including anti-insecticidal, anti-acaricidal, anti-nematicidal, anti-bacterial, anti-fungal, and anti-viral properties. Research studies have reported excellent antibacterial activity against Gram-positive and Gram-negative multidrug-resistant bacteria as well as methicillin-resistant Staphylococcus aureus (MRSA). Although there has been an extensive investigation on the antimicrobial properties of *Cannabis*, the antimicrobial properties of *Cannabis* on phytopathogens and aquatic animal pathogens, mostly those affecting fish, remain under-researched. Therefore, the current review intends to investigate the existing body of research on metabolomic profile and anti-microbial properties whilst trying to expand the scope of the properties of the *Cannabis* plant to benefit the health of other animal species and plant crops, particularly in agriculture.

## 1. Introduction

*Cannabis* (sativa, indica, and ruderalis), which belongs to the family Cannabaceae, is the most ancient, domesticated crop. It is well known for its various medicinal properties and recreational purposes [1,2]. The positive effects of the medicinal use of marijuana have been used to treat conditions such as chronic pain, multiple sclerosis, nausea, vomiting, terminal diseases, and epilepsy. These effects have been known for many years in various parts of the world, even before scientific research into its effects [3]. According to Cascini et al. [4], *Cannabis* has been reported as the most consistently misused plant in many parts of the world, including South Africa. This was probably due to a lack of in-depth research on the healthful properties and the negative connotations attached to *Cannabis*, thus inhibiting efforts to expand the knowledge about the rich spectrum of substances formed by the plant. Currently, the *Cannabis* plant (Figure 1) is increasingly exploited worldwide for several agricultural and industrial applications [5].

The metabolomic profile of *Cannabis* has thus far proven to unveil a large variety of bioactive metabolites or phytochemicals [6] that are produced by different plant parts, such as stem barks, leaves, flowers, and buds. Previous reports suggest that there are approximately more than 700 strains of *Cannabis* with a few suggested to possess powerful medicinal properties [7]. However, when the main focus is on the chemical composition of numerous *Cannabis* strains or cultivars, the chemical classification of the strains is limited to several categories that are mostly based on the plant’s cannabinoid content. Furthermore, there are fewer research studies aimed at the discovery of new phytochemicals of various *Cannabis* strains that may possess therapeutic or medicinal properties.

The discovery of *Cannabis*-derived compounds has been frequently reported to be hugely linked to their pharmacological properties, such as their psychoactive, anti-inflammatory, antimicrobial, and antioxidant properties [8]. In addition, the promising effect of being an analgesic for neuropathic pain has also been observed by Fraguas-Sánchez and Torres-Suárez [3]. This pharmacological activity of *Cannabis* has predominantly been attributed to the presence of cannabinoids and essential oils as reported by Nissen et al. [9]. Despite a long history of medical use, the therapeutic potential of Cannabis roots and stem barks has been largely ignored in modern medical research and practice.

In addition, although there has been an effort to relax *Cannabis* legislation in many countries, there is still a lack of investigative studies that focus on assessing the effects of different environmental conditions on the production of bioactive phytochemicals, particularly in *Cannabis* indica and some other *Cannabis* strains. In contrast, *Cannabis sativa* remains the only cultivar that has dominated the research [10]. Similarly, among the profiled secondary metabolites with promising therapeutic effects, cannabinoids, terpenes, terpenoids, or flavonoids have been at the center of medicinal research and have been substantially explored, especially for their antimicrobial potential [11]. To date, there is still a pool of *Cannabis*-derived compounds that have not yet been investigated for their anti-microbial properties. These untested compounds may serve as a good source of potential antimicrobial agents with novel modes of mechanisms against various pathogens. Regardless of several studies having extensively investigated the antimicrobial properties of *Cannabis*, the antimicrobial properties of *Cannabis* on phytopathogens and aquatic vertebrate animal pathogens, especially fish pathogens, remain largely unexplored.

In many countries, research on the effects and properties of *Cannabis* could only be carried out in recent years and has focused more on several human microbial pathogens due to their distinctive activities, thus neglecting many other pathogens causing serious economic loss in agriculture. As many claims suggest that *Cannabis* possesses low toxicity, it could be the best candidate for the control of aquatic pathogens and phytopathogens as an alternative to synthetic chemicals. Claims abound that extracts from the dried leaves and flowers of *Cannabis* (often used in a variety of formulations) possess repellent and killing potential against various crop pests [12]. For instance, pure cannabinoids are reported to kill or repel mites, insects, nematodes, fungi, and protozoa [13]. However, empirical evidence on specific crop pathogens that possess this effect and the non-phytotoxic concentration of these cannabinoids or *Cannabis* extracts on the plant growth of various crops is often not determined by the computer-generated concept of mean concentration stimulation point (MCSP)—the concentration that would not induce phytotoxicity in the protected plant species while suppressing population densities of crop pathogens [14]. Moreover, most of the scientific literature describes in vitro experiments, and few studies focus on fieldwork. The protection of crop plants from phytopathogens through plant extracts is another form of biocontrol that has long emerged as an alternative to synthetic chemicals. This method has proven to play a pivotal role resulting in better cost-effectiveness as reported by Shricharan et al. [15] given the high cost of synthetic chemicals and their detrimental effect on the environment and human health. Many studies have shown that nearly all types of *Cannabis* tissues exhibit antimicrobial activity, including antifungal and antiviral activity. Nevertheless, remarkably, this antimicrobial activity is not as extensively studied and as strongly pronounced as the antibacterial activity. Some published papers have reported that extracts of *Cannabis indica* leaves, seeds, and stems possess significant antifungal activities against *Aspergillus niger*, *Aspergillus parasiticus*, and *Aspergillus oryzae* [16]. In addition, another study by Gyawali et al. [17] revealed excellent antibacterial activity against Gram-positive and Gram-negative multidrug-resistant bacteria as well as methicillin-resistant Staphylococcus aureus (MRSA).

*Cannabis* seed oil has been shown to possess notable antibacterial activity against *Bacillus subtilis* and *S*. *aureus* and moderate activity against *Escherichia coli* and *Pseudomonas aeruginosa* [18]. A recent research study discovered the high potency and efficacy of *Cannabis* inflorescence and root extract against American Foulbrood caused by *Paenibacillus* larvae, with low toxicity to honeybee larvae [19]. Terpenes are reported to have promising antimicrobial properties against the most investigated methicillin-resistant *S*. *aureus*, a Gram-positive bacterium [20]. Another recent study has discovered that an extract of *Cannabis* leaves was able to effectively inhibit the growth and development of *E*. *coli* and *klebseila* spp. isolated from the *Oreochromis mossambicus* [21]. In addition, previous studies have extensively highlighted the antimicrobial properties in the literature, and *Cannabis* has a substantially unexploited anti-viral potential against different human viral pathogens. 

Based on previous research studies, most medicinal plants with substantial antiviral activity, as well as those containing new plant-derived antiviral compounds, have been found to treat viral infections in people and animals [22,23]. Recently, scientists found that the anti-inflammatory properties of cannabinoids are associated with the treatment of COVID-19, the virus infection that ravaged almost the entire world [24]. Phytocannabinoids, notably cannabidiol (CBD), tetrahydrocannabinol (THC), and terpenes, have shown the potential to reduce severe acute respiratory syndrome coronavirus-2 (SARS-CoV-2) viral infection by downregulating Angiotensin-converting enzyme 2 (ACE2) transcript levels [25]. These compounds have also been reported to act as the main protease inhibitors that block viral replication and exert an immunomodulatory effect by controlling the excessive release of cytokines, which is commonly associated with SARS-CoV-2 infections [26]. 

Apart from cannabinoids, terpenes in *Cannabis* plants have also been widely explored for their antiviral properties, and studies including other unique secondary metabolites have been limited. Concurrently, regardless of the historical and extensive research evidence, the antiviral properties of *Cannabis* against viruses affecting livestock animals and poultry have been understudied as an alternative, given that antiviral drugs and veterinary medicines are limited. This is due to viral mutant resistance to existing antivirals, emerging new viral pathogens, toxic side effects, and high costs [27]. Therefore, the focus of this review is to systematically review the scientific literature and establish the existing research and scientific data about the metabolomic profile of *Cannabis* and its inhibitory effect on pathogens whilst identifying the research gaps that still need further investigation. Furthermore, this review article is intended to foster the possible groundbreaking discovery of other metabolites. 

## 2. Materials and Methods

The systematic review produced was based on relevance to the topics of the profile of *Cannabis* and its anti-pathogenic properties. We followed PRISMA (Preferred Reporting Items for Systematic Reviews and Meta-Analyses) [28] as a guide for the completion of this systematic review. The full search is indicated in Figure 2 below:

### 2.1. Search Strategies 

A systematic literature search was carried out by consulting five electronic scientific databases: PubMed, Scopus, Science Direct, Semantic Scholar, and Google Scholar. A combination of the following keywords was used: “*Cannabis*”, “metabolomic profile” “phytocannabinoids”, “antimicrobial properties”, antiviral properties”, “antibacterial properties”, and “antifungal properties”. The publication year was used as a restriction to the retrieved articles from 2010 through April 2023 and included journal articles, review papers, and research reports published in English only. 

### 2.2. Selection Criteria 

The selection criteria were based on the PRISMA statement [28]. The search mainly focused on the original research papers published from 2010 until April 2023 papers that evaluate the analysis of *Cannabis* plant materials and bioactive compounds as the subject of interest. There were no limits for *Cannabis* plants: (herbal form—the leaves, flowering tops, and resin form—hashish, or hash oil), varieties (*indica*, *sativa*, or *ruderalis*), and gender (male, female, or monoecious). The search span was from the years 2010 to 2023. All articles before 2010 were excluded from the search. 

### 2.3. Risk of Bias Assessment

The study is based only on original research articles, review papers, and conference papers. To maintain the quality of the review, all duplications were checked thoroughly. Abstracts of the articles were included in the review process to ensure the quality and relevance of the academic literature. A careful evaluation of each research paper was carried out at a later stage.

## 3. Results and Discussion

### 3.1. The Use of Advanced Metabolomics Tools in Research Studies

Metabolomics is a multidisciplinary scientific study involving the identification and classification of all small molecules, known as metabolites in a biological system [29]. Most importantly, when metabolomics is used in conjunction with genomics, transcriptomics, and proteomics, it helps shed light on the workings of biological systems as they develop and respond to environmental stimuli and illustrate the complex connection between different cellular events [30]. Most research studies relied on the application of metabolomics to primarily answer biological questions that are otherwise unattainable using conventional methods. According to Maia et al. [31], metabolomics is considered a bioanalytical tool that is largely employed in medicinal plants due to its potential to conclusively allow the characterization of various metabolites present in a single extract. Due to the versatility and unique capabilities in the deconvolution of the metabolite composition of complex matrices, most research studies in medicinal plants and drug discovery have been substantially accelerated [32]. As a multidisciplinary science, metabolomics has outmaneuvered traditional boundaries of scientific ventures, contributing significantly to molding data-driven discoveries and knowledge generation trajectories [33]. Recent studies conducted in metabolomics profile untargeted and targeted primary and secondary metabolites, and analyzing molecular studies has proven that metabolomics has outsmarted and bypassed the “normal” nutritional and growth-related studies usually conducted in Africa. To date, metabolomics has been developed for a wide range of applications in various fields, such as plant [34] and food science [35,36,37,38], medicine [39], toxicology [40,41], environmental sciences [42], plant protection products (PPPs), R&D [43,44], and soil science [45]. Since its introduction to the above fields, it has proven to be a crucial approach for gaining insight into the largest possible set of low-molecular-weight metabolites present in a biological sample and has been reported to be an effective tool to unearth and solve research questions. It should be noted that metabolomics has been used immensely in developed countries, but most African countries, except a few plant and soil studies conducted in South Africa, are still lagging in adopting advanced and unconventional methods to solve research questions. In South Africa, there is only a small number of plants (agricultural and medicinal) to which this advanced technology has been applied to unearth difficult research questions. Therefore, future studies should incorporate metabolomics in many other plant-natural products and research fields to highlight the several metabolites that are yet to be explored. 

### 3.2. Application of Metabolomics in Cannabis (Cannabinomics)

According to [46,47,48], nuclear magnetic resonance and mass spectrometry MS-based [49,50,51,52] analyzers are the main analytical platforms used in metabolomics analysis. In addition, analyzers equipped with triple quadrupole (QQQ) detectors, such as LC/QQQ/MS and GC/QQQ/MS systems, are crucial in *Cannabis* research due to their superior selectivity and sensitivity in quantitative analyses [53,54]. However, in the case of *Cannabis*-derived matrices reported to have high complex metabolome, employing various analysis tools is highly recommended as reported by [5]. The use and exploitation of *Cannabis* have sparked controversy; however, the recent legalization of its use for medical and other purposes in many countries within the corresponding legislative framework [1,55], in combination with the remarkable bioactivities of the plant, pose an urge for the acceleration and intensification of *Cannabis* research and development. Studies by [55,56] revealed that the application of cannabinomics in mapping the metabolome could assist a great deal in *Cannabis* research and development. More studies in cannabinomics will also aid in understanding the growth and development of *Cannabis* and the factors (light, water requirements, and growing conditions) that can contribute to a greater yield of *Cannabis*. Simultaneously, the abovementioned is likely to have a significant impact on drug discovery, medicine, food science, functional cosmetics research, and metabolic engineering of microorganisms for the biosynthesis of cannabinoids as reported by [57]. Although most studies on *Cannabis* have been on flowers and oils, there has been an interest in researching the level of cannabinoids and terpenoids in edibles, *Cannabis* sweets, medicine, and cosmetics as indicated by [58,59]. 

### 3.3. Cannabis Active Compounds

It is reported that *Cannabis* is one of the medicinal plants that possess a high metabolomic profile with a diverse range of essential bioactive compounds used for numerous pharmacological purposes [60]. For instance, a previous study by Fischedick et al. [61] revealed that more than 600 compounds have been discovered, with 180 belonging to the cannabinoids family and including many terpenes. In addition, Peng and Shahidi [10] indicated that their biosynthesis involves the alkylation of olivetolic acid with geranyl diphosphate. A recent study by Jin et al. [62] reported that the abovementioned reaction leads to the formation of cannabigerolic acid CBGA as shown in Figure 3, which is a precursor molecule for numerous other cannabinoids, such as ∆^9^-tetrahydrocannabinol (∆^9^-THC), which is regarded as the most important natural cannabinoid [63]; cannabidiol (CBD); cannabigerol (CBG), cannabinol (CBN); cannabichromene (CBC) [63]; Δ^8^-tetrahydrocannabinol (Δ^8^-THC) [64]; Δ^9^-tetrahydrocannabivarin (THCV) [64]; cannabivarin (CBV) [64]; and cannabidivarin (CBDV) [64]. Some researchers discovered the highest amount of THC in female inflorescences of *Cannabis* [65]. Another recent research study discovered that biologically active cannabinoids such as Δ^9^-THC, (25.04%) and CBD (resorcinol, 2-p-mentha-1,8-dien-4-yl-5-pentyl) (50.08%) were found in *Cannabis* resin in a high percentage [66]. 

Cannabinoids are the best-known secondary active metabolites reported extensively in modern and scientific literature due to a considerable number of studies demonstrating their effectiveness in the treatment of some infections caused by a wide range of pathogens [67]. Consequently, this is a reaffirmation of many previous research studies that asserted that cannabinoids are usually concentrated in the viscous resin produced in structures known as glandular trichomes. These findings prove that the presence of biologically active phytochemicals in high concentrations makes it a valuable source to be used in herbal preparation for different ailments. However, attention has mostly been given to cannabinoids resulting in many other compounds, such as nitrogen compounds, amino acids, hydrocarbons, carbohydrates, organics, and fatty acids, not being thoroughly investigated for their medicinal potential. In addition, the full potential of *Cannabis* has not been sufficiently exploited because some active molecules have not been defined and their cellular and molecular mechanisms that underlie its antimicrobial and anti-inflammatory activity are not investigated nor understood in depth. However, it is noteworthy to mention that beyond *Cannabis*, several plants have been reported to possess cannabinoid-like active compounds that interact with the cannabinoid system in one way or the other. For instance, studies conducted have reported the roots of *Otanthus maritimus* L. to be the main receptors of CB1 and CB2 of cannabinoids [68,69]. In addition, Ruta graveolens, which is used in traditional medicine for its healing properties as fresh herbs, infusions, decoctions, powders, or oils [70,71], was found to have a selective affinity to the CB2 receptors as reported by Wu et al. [72]. A recent study by Dimmito et al. [73] revealed carrots (*Daucus carota*, L) contain a high concentration of falcarinol (FaOH), which acts as a moderate skin irritant, aggravating histamine-induced edema skin, and showed receptor affinity to both CB1 and CB2 with a CB1 antagonist profile [74]. Catechins and epi-catechin compounds in tea (*Camellia sinensis* L.) were found to be receptors of CB1 [75]. Furthermore, Kava Kava (*Piper methysticum forster*) also known as “intoxicating pepper”, which is used as a traditional medicine to cure pains, headaches, convulsions, menstrual pains, and skin diseases [76], was reported to have the ability to activate the CB1 and CB2 receptors [77]. Some phytocannabinoids have been discovered in flowering plants, such as liverworts which are known to be prolific producers of compounds with a bibenzyl backbone as reported by Chicca et al. [78]. Furthermore, Yang et al. [79] conducted a study on several *Rhododendron* species (Ericaceae) which proved to produce bioactive meroterpenoids with a cannabinoid backbone which are reported to be closely associated with the CBC type decorated with an orcinol side chain. Other studies by Fuhr et al. and Pollastro et al. [80,81] showed that the edible roots of *Glycyrrhiza foetida Desf*. (licorice; Fabaceae) and *Amorpha fruticosa* L. (bastard indigo; Fabaceae) were found to contain amorfrutins—bioactive compounds with a cannabinoid backbone. 

To date, about 400 of the chemical substances found in *Cannabis* plants are terpenes and phenolic compounds [82]. Terpenes are a chemical class of compounds acclaimed for their antioxidant, anti-inflammatory, and antimicrobial properties [83]. However, most primary, and secondary metabolites derived from *Cannabis* have not been sufficiently explored and studied. Therefore, an extensive investigation of other unique compounds should be extensively investigated for possible antimicrobial properties as it is becoming clear that a much wider range of *Cannabis* constituents may be involved in various therapeutic and/or medicinal effects. According to Ryz et al. [84], other active compounds that have been identified in *Cannabis* roots include triterpenoids friedelin, epifriedelanol, and the sterols β-sitosterol, stigmasterol, and campesterol (Figure 4), but neither cannabinoids such as THC nor CBD have been reported to be present. Triterpenoids and sterols have also been identified in stem barks [85]. 

Recent findings revealed that these compounds, especially epifriedelanol, possess anticancer, anti-inflammatory, and antisenescence activities [86]; whereas, triterpenoids friedelin has been reported to display a wide spectrum of anti-inflammatory, antioxidant, antipyretic, anticarcinogenic, and antitumor effects in a variety of plants [87]. In addition, a recent study reported that sterols β-sitosterol has been shown to inhibit aromatase and 5-alpha-reductase, an activity exploited to treat pathologies such as benign prostatic hyperplasia and androgenetic alopecia [88]. Despite the historical claims of the medicinal activity of bioactive compounds in *Cannabis* roots, their therapeutic potential continues to be largely ignored in modern times. Currently, no research study has investigated the antimicrobial activity of friedelin or epifriedelanol and ST against antimicrobial activity (pathogenic bacteria, fungi, viruses, and nematodes) specifically isolated from *Cannabis* roots and stem barks. Furthermore, no information is available regarding their cellular and molecular mechanisms determined through preclinical and clinical studies. Certainly, this suggests the need for the re-examination of whole root preparations on microbes employing modern scientific techniques to provide historical and ethnobotanical claims of clinical efficacy.

### 3.4. Metabolomics Pathways of Cannabinoid

Cannabinoids were the first identified group of potent *Cannabis* metabolites, with the medicinal properties of its major representatives being attributed to their interference with the G protein-coupled cannabinoid receptors [89,90]. According to Hanuš et al. [91], about 150 cannabinoids have been identified in hemp. The two biosynthetic pathways include the polyketide pathway, which leads to olivetolic (OLA), and the plastidial 2-C-methyl-D-erytritol 4-phosphate (MEP) pathway, which leads to geranyl diphosphate (GPP). Precursors OLA and GPP form cannabigerolic acid (CBG-A), which is a precursor for different cannabinoids, as well as THC-A, CBD-A, and CBC-A as reported by Andre et al. [92]. Olivetolic is a key enzyme in the cannabinoid biosynthetic pathway as revealed by Morita et al. [93]. Studies by Carvalho et al. and Luo et al. [5,94] reported on the recent development in the biosynthesis of cannabinoids by genetically engineered organisms, which could potentially provide solutions to the large-scale production of rare cannabinoids. 

Since *Cannabis* is an open-pollinated plant, it is prone to no-uniformity [50]. A study by Small and Antle [95] revealed that the reasons for higher heterogeneity in *Cannabis* varieties could be because pollen can disperse a few kilometers in relation to the wind direction. In addition, a recent study by Eržen et al. [96] reported that in *Cannabis,* the varieties (strains) are often not fully inbred; therefore, they have a relatively high level of heterogeneity and instability, compared to other crops. 

### 3.5. Antibacterial Activity of Cannabis 

Generally, the antibacterial activities of *Cannabis* against various resistant bacterial pathogens have been extensively studied, particularly against human pathogens [97]. Despite the substantial investigations undertaken about antibacterial activities, most research studies focused on the response of the same pathogenic bacteria such as *Bacillus* or *Staphlococcus* [98,99]. In addition, while inflorescence has been empirically confirmed to contain the highest concentration of active compounds, surprisingly, most studies assessing the antibacterial properties of *Cannabis* leaves and seeds receive much more attention when preparing extracts for their antibacterial properties [100]. The chemical characteristics of many of these active compounds, such as THC, CBD, CBN, cannabivarin (CBV), tetrahydrocannabivarin (THV), and many other compounds such as fatty acids, have been reported to exert membrane disruption on bacterial species [101]). While most studies outlined the efficacy of active compounds against several clinically relevant bacterial strains, mechanisms of membrane disruption and the potential application of these active compounds as a therapeutic against certain pathogenic bacteria is not clear nor documented [102]. Although many of these active compounds have been shown to be effective antibacterial agents, little is known about the possibility of their synergistic relationships with antibiotics against bacterial infections. Therefore, the aspect of therapeutic design such as the utilization of co-therapies whose efficacy is conferred through synergistic relationships between two or more therapeutic agents holds promising results in the development of innovative antibacterial agents [103]. 

Many extracts of various *Cannabis* parts, such as roots and leaves, have also been extensively researched for their antibacterial properties. Isahq et al. [16] conducted an in vitro research study and discovered that *C. indica* (leaves, stems, and seeds extracts) possess strong anti-bacterial activity against multidrug-resistant bacterial strains such as *S. aureus*, *Bacillus cereus*, *E. coli*, *Klebsiella pneumoniae*, *Pseudomonas aeruginosa*, and *Proteus mirabilis*. In addition, there is a growing body of evidence indicating that multiple cannabinoids administered at 2 µg/mL are potent inhibitors of *S. aureus* biofilm formation, with cannabigerol (CBG) at 4 µg/mL seemingly able to disperse preformed biofilms and rapidly kill persister cells [104]. Moreover, phytocannabinoids such as cannabichromenic acid (CBCA) have proven to exert potent antibacterial activity against clinical strains of *Enterobacter faecalis* and both methicillin-resistant and sensitive strains of *S. aureus* [105]. Another study by Marini et al. [106] investigated the antimicrobial properties of “hashish” against common hospital-associated bacterial strains and found that *Cannabis* extract exerted the greatest antimicrobial effects on *S. aureus* 25923, with an inhibition zone of 14 mm, followed by MRSA, *E. coli* and *Klebsiella pneumoniae* with values of 12, 10, and 7 mm, respectively. On the other hand, essential oils of many plants have been shown to contain a wide series of secondary metabolites that can inhibit or slow the growth of pathogenic bacteria [107].

Some previous research studies reported a strong antibacterial activity exerted by seed oil (21–28 mm) against *Bacillus subtilis* and *S. aureus* and moderate activity (15 mm) against *E. coli* and *P. aeruginosa* (16 mm) [18]. Recently, when the efficacy of the essential oils from *Cannabis*, especially the Futura variety, was evaluated as an antimicrobial agent, it showed promising results. [108] have reported that *Cannabis* essential oils proved to reduce the virulence of the food contaminant *Listeria monocytogenes*, with a possible application in the food-processing industry. Furthermore, the inhibitory activity of *C. sativa* seed extracts was observed in *S. aureus* biofilm formation, which indicates that these extracts could have enormous potential as preservatives in both the food and cosmetics industries [109]. More recently, cannabinoids were found to be more effective in reducing the bacterial colony count in dental plaque when compared with commercial toothpaste such as Oral B and Colgate, which suggests that *C*. *sativa*-derived compounds, especially CBD, could be used for oral care applications [110]. Therefore, further extensive screening of *Cannabis* essential oils from different varieties is necessary and should be undertaken to identify the samples with the most promising biological value. 

Nasrullah et al. [109] assessed the in vitro antimicrobial activity of methanol and n-hexane extracts of *C. sativa* leaves against *Bacillus cereus*, *B. subtilis*, *E. coli*, *P. aeruginosa*, and *Salmonella* species. Methanol extract revealed better antibacterial activity than the n-hexane extract. Other, studies discovered that crude alkaloids extracted from *Cannabis* leaf present antibacterial effectiveness against bacterial strains representative of skin, mouth, and ear microflora and against the β strain of *E. coli* [110]. 

### 3.6. Antifungal Activity of Cannabis 

The continuous emerging empirical evidence regarding the antifungal properties of *Cannabis* and its secondary metabolites has proven to present excellent results. A few available research studies suggest that, for a long time, *Cannabis* has been regarded as possessing antifungal activity against a diverse range of pathogenic fungi [61]. Despite cannabinoids such as Δ^9^-THC and CBD being known to greatly contribute to antifungal activity, some studies have demonstrated that plant extracts or essential oils also present this activity [9,111]. So far, studies about the antifungal activity of *Cannabis* extracts and their secondary metabolites are extremely limited as compared with research studies on antibacterial activities [112]. Wanas et al. [113] showed that the n-hexane fraction of *C. sativa* effectively inhibits the growth and development of a life-threatening human pathogen *Cryptococcus neoformans* which is responsible for lung infections in humans. The results revealed modest antifungal activity with an IC50 value of 33.15 µg/mL against *C. neoformans*. Other findings revealed that *Cannabis* petroleum ether extract has effectively displayed modest activity against *Candida albicans* [114].

Some recent studies suggest that a 6.25 mg mL^−1^ n-butanol leaf extract of *C. sativa* has proven to completely control the growth of *Aspergillus versicolor* [115]. Furthermore, a significantly high percentage of mycelial growth inhibition was observed in seed-borne phytopathogenic fungi *Alternaria* species [116]. Despite numerous research studies confirming the antifungal activities of *Cannabis*, most studies investigated mostly extracts from its aerial. Studies with the extracts of roots and stems have not been extensively carried out. Therefore, it is highly probable that extracts from roots and stems possess pronounced antifungal potential against a wide spectrum of resistant pathogenic fungi that cause devastating economic loss in agriculture. 

More recently, there has been growing scientific evidence suggesting that the synergistic interaction of *Cannabis* compounds with common antibiotics has the potential to exert antimicrobial action on resistant pathogens [117]. Some previous studies have demonstrated that the synergistic effect between cannabinoids and terpenes may contribute to more effective treatment [118]. However, this synergistic interaction effect is not as extensively investigated in antifungal and cytotoxic activities. In addition, this knowledge appears to have not been given sufficient attention, especially in its application in medicine, industry, or agriculture. Moreover, it is widely known that since the emergence of diseases in fish farming, the use of herbal products to control fish pathogens in aquaculture is an alternative and current practice as chemicals or drugs used to treat infected fish are limited and have harmful effects to the environment and humans who are end consumers of fish products.

Some previous research studies observed that various plant extracts and essential oils can inhibit the growth or even cause the death of pathogenic fish fungus at comparatively low concentrations [119,120]. For instance, *Origanum onites* and *Thymbra* have been observed to possess potent antifungal properties against *Saprolegnia parasitica* known to cause saprolegniasis diseases and considerable economic problems on hatched salmon in aquaculture facilities [121]. In addition, Terminalia *catappa* has been reported to exhibit the highest antimicrobial effect of minimum inhibitory concentration (MIC) of 25 and 12.5 mg mL^−1^ against *S. parasitica* [122]. However, currently, no scientific reports have been published concerning the antifungal potential of *Cannabis* against fish fungal pathogens and there is also no investigation on the possible synergistic interaction between standard antimicrobial drugs and *Cannabis* compounds. Consequently, the urgent identification of new molecules exerting novel modes of action that can work alone or in combination with synergistic compounds is required [20].

### 3.7. Antiviral Activity of Cannabis

One of the recent research studies shows the possibility of *Cannabis*-derived compounds possessing antiviral properties with the potential to reduce morbidities and slow down the progression of various types of viral illnesses [123]. However, the antiviral properties of *Cannabis* are the most understudied in modern medical research. Although *Cannabis* has proven to synthesize a wide range of bioactive compounds, only cannabinoid compounds, particularly THC and CBD, appear to have been explored for their antiviral potential, and there exists insufficient knowledge on the antiviral action modes of cannabinoids on various types of viruses. According to DeMarino et al. [124], CBD is known for its ability to induce apoptosis in mammalian cells and has been demonstrated to reduce extracellular vesicle release from HIV-infected monocytic cells and their viral cargo. Lowe et al. [125] assessed the antiviral effects of CBD in hepatitis B and hepatitis C and discovered that it was not active against HBV infection in vitro but exerted a cytotoxicity effect on the liver cell line, which was used to culture the virus.

In another clinical trial, *Cannabis* treatment was associated with lower levels of pro-inflammatory biomarkers in cerebral fluid (CSF) of HIV patients [126]; whereas, some studies reported that the treatment of Kaposi sarcoma-associated herpesvirus (KSHV) with various concentrations of CBD significantly reduced VEGFR-3 levels, suggesting another mechanism whereby CBD may affect the proliferation and viability of Kaposi sarcoma through the VEGFR-3 signaling pathway [127]. Similarly, in an HCV assay, CBD was reported to have successfully inhibited the virus by 84.5% with minimal cytotoxicity effect. Therefore, the direct antiviral activity of CBD towards HCV has demonstrated that the compound affects hepatitis, which is caused by activated T cells and macrophages as reported by Lowe et al. [125]. A previous research study by Adejumo et al. [128] reported that the use of *Cannabis* is related to a lower incidence of liver cirrhosis in chronic HCV patients. Further, longitudinal studies are still needed to ensure the effects of *Cannabis* extracts and bioactive compounds to verify their effectiveness.

Scientists have recently observed cannabinoids, especially THC and CBD, as possible drugs for SARS-CoV-2 [129]. Reviews and investigations on SARS-CoV-2 are currently dominating the research, resulting in fewer studies on some viruses or the possible re-emergence of other viruses. This is primarily due to empirical evidence revealing that cannabinoids such as CBD may be an efficient inhibitor of SARS-CoV-2 (strain 229E) replication in human lung fibroblasts (MRC-5) through enhancement of antiviral terpene efficacy [130]. In addition, cannabidiolic acid (CBDA) and cannabigerolic acid (CBGA) have been proven to interact with the SARS-CoV-2 spike protein S1 subunit and prevent the entry of several live viral variants into human epithelial cells [131]. The mechanism of action of cannabinoids in SARS-CoV-2 inhibition has been reported to include inhibition of viral cell entry, inhibition of viral proteases, and stimulation of cellular innate immune responses [129]. There are also reports of CBD’s ability to inhibit TMPRSS2 in several models of human epithelia. Additionally, CBD and THC, acting as CB2 receptor agonists, reduce the level of pro-inflammatory cytokines in lung cells [123]. 

Other recent studies have shown the ability of ∆^9^-THC to inhibit viral 3CL pro caused by SARS-CoV-2 at IC50 3.62 μM [132] while others demonstrated a promising effect of ∆^9^-THCA-A in molecular modeling, thus, predicting inhibition of human angiotensin-converting enzyme 2 (ACE2) [133]. CBGA has been noted to bind orthosterically and allosterically to the SARS-CoV-2 spike protein S1subunit (Kd = 5.6 μM) and prevent the cell entry of human epithelial cells and Vero cells by SARS-CoV-2 and early variants [131].

Moreover, it has been proven that cannabinoids can effectively produce a heightened pharmacological effect than individual isolated compounds by working synergistically with other specialized metabolites via the “entourage effect” [134,135]. It is, however, worth noting that the interaction of cannabinoids with other bioactive metabolites against the viruses has not been adequately studied nor investigated. Concurrently, there exist no reported studies on the antiviral effects of *Cannabis* plant extracts (roots, leaves, flowers, and seeds) and essential oils. Therefore, further studies should be conducted in the quest to find a therapeutic agent that will provide alternative treatments to prompt sensitive, cost-effective management of viruses. Furthermore, whilst it is widely known that infectious viral diseases remain an important problem all over the world, most research studies have focused on the viruses affecting humans, thus de-emphasizing research on the antiviral potential of *Cannabis* against livestock animals and crop viruses. Many studies have begun to look at the possible use of feed additives (FA) such as essential oils, extracts, and by-products from medicinal plants. These could be included in an animal’s diet to control pathogens [136] at the molecular level. In addition, the use of feed additives, especially in South Africa, is still limited to experiments, and there is also a lack of application at the farm level. Furthermore, a systematic review of the literature retrieved little information on the use of *Cannabis* feed additives in fish farming. Consequently, thorough investigations concerning the tolerance or effects of graded levels of *Cannabis*-derived compounds, especially CBD and THC, in animals, especially livestock, should be undertaken to ensure their safety as there are currently less well-controlled studies, most of which focus on companion animals and poultry [137]. Research by Mahmoudi et al. [138] on poultry demonstrated significant changes in blood parameters, particularly cholesterol levels. 

### 3.8. Anti-Nematicidal Activities of Cannabis 

The ability of plant-derived extracts and compounds to either paralyze or kill nematodes has spurred more interest in the discovery of plants that can be used as control agents for parasitic nematodes, and thus far, studies relating to this have received substantial attention [139,140]. This use of plant extracts and compounds is already being greatly exploited commercially in pest management and a rising trend towards organic farming [141]. This is because several pesticides and nematicides are linked to human health and environmental hazards, which have become a worrisome public health concern [142]. This has propelled a need to find alternative methods that are safe and suitable for the environment [143]. Extracts of various plants and pure cannabinoids contained in *Cannabis* have been found to possess nematicidal properties and have proven to be effective for the control of a wide range of parasitic nematodes [144]. Some previous research demonstrated a significant reduction in several galls, egg masses, nematode fecundity, and build-up caused by *C. sativa*, and the maximum reductions in these variables were recorded with a 20 g dosage [145].

Similarly, some research studies confirmed that the use of dried flowers and leaves of *Cannabis* is effective in killing or repelling plant pathogenic nematodes [146]. However, the current body of knowledge on the investigation of *Cannabis* against pathogenic nematodes remains quite limited. In addition, this is probably because few researchers have focused only on phytopathogenic nematodes, most notably root-knot nematodes, and thus far, there are no studies that have investigated the nematicidal properties of *Cannabis* against nematodes causing diseases in humans. Although many plants are recommended for use against gastrointestinal nematodes, especially in animals, and related problems in many parts of the world, it is surprising that, to date, no studies have investigated the medicinal activity of *Cannabis* against gastrointestinal nematodes affecting livestock animals. Consequently, questions on the role of psychoactive, medicinal, or other constituents in human nematode enhancement or suppression also must greatly be addressed in experimental studies.

Most publications on *Cannabis* nematicidal properties usually are reports involving leaves and flowers as extracts and provide little or no useful data on the other parts of the *Cannabis* plant, such as roots, stem, and seeds [147]. Despite the little information on the nematicidal properties of *Cannabis,* the continuous display of in vitro nematicidal activity by other plants, such as *Eucalyptus globulus*, inconsistent suppression of the *Meloidogyne incognita* population on tomato roots is evidence enough that a lot more work still needs to be completed in relation to nematicide properties of *Cannabis* given its copious bioactive metabolites [100]. Likewise, the essential oils of *Eucalyptus citriodora* and *Pelargonium asperum* L. proved to be nematicidal against *M. incognita* [148]. Some recent studies revealed that gall formation and multiplication of both *M. incognita* and *M. hapla* on tomato roots have been found to have been significantly reduced by all soil treatments with *Artemisia annua* meal powder and water extract [149]. On the other hand, a study by Bawa et al. [150] conducted on the neem plant (*Azadirachta indica*), red-bell pepper (*Capsicum annuum*), ginger (*Zingiber officinale*), and African locust bean has shown to have completely prevented attack and hatching of *M. incognita* eggs and destroyed 100% of the juveniles at 1000 ppm concentrations (at 10% and above). Leela et al. [142] reported that aqueous leaf extracts of *Strychnos nuxvomica* caused 100% mortality of second-stage juveniles of *M. incognita* at a 2% concentration. Another study reported that extracts from *Nicotiana tabacum* and *Acorus calamus* were found more effective in killing *M. incognita*, with an EC50 that was 5–10 times lower than the EC50 of the synthetic pesticides chlorpyrifos, carbosulfan, and deltamethrin [151]. This, therefore, suggests that nematicidal effects may be more potent in plant extracts than synthetic nematicides. 

### 3.9. Acaricidal Activities of Cannabis

Since a large number of research studies confirmed various plant-derived compounds possessing powerful pharmacological potential, few researchers investigated the acaricidal properties of *Cannabis* and its secondary metabolites. However, a systematic review of the literature retrieved extremely little information on the therapeutic use of *Cannabis* against parasitic mites and ticks. These mites and ticks classified in a taxon of arachnids known as the acari have been frequently reported to cause deleterious effects on humans, animal health, and crops, subsequently resulting in significant economic losses [152]. For instance, ticks have been observed to economically impact cattle production by reducing weight gain and milk production [153]. In addition, mites are greatly reported as a major pest for attacking crops of worldwide economic importance and numerous agricultural production systems [154].

For a very long time, *Cannabis* has been traditionally employed as a repellent and pesticide to protect crops from phytophagous arthropods without an in-depth knowledge of the bioactive compounds possessing this effect, and most publications were anecdotal [155]. A few research investigations on selected crops have been tested, predominantly on terpenoids which constitute the plant’s essential oil [156]. Several other classes of natural products known to be minor constituents of *Cannabis*, such as flavonoids, phenols, polyphenols, phytosterols, amines, lignanamides, and fatty acids, are poorly investigated regarding acaricidal properties, and only a few have been exploited commercially [156]. Although in human medicine, the focus has always been on phytocannabinoids, some papers reported that a great number of highly volatile terpenoids from *Cannabis* have the potential to act as an insect repellent and antifeedant against grazing animals [157]. 

A recent study by Nasreen et al. [158] investigated the efficacy of *C. sativa* against *Rhipicephalus microplus* ticks and the high lethal effect of *C*. *sativa* on egg laying with an index of egg laying of 0.26 and 0.24, respectively, egg hatching (33.5 and 37.1, respectively), and total larval mortality (100%, both), at 40 mg/mL. In addition, when this was applied to cattle that had been inoculated with larvae ticks, it was observed that a 45% solution of both herbal extracts significantly reduced the number of ticks by 96 h post-treatment. Several terpenoids and cannabinoids of *Cannabis* also propelled the investigation of acaricidal effects against *Dermanyssus gallinae* and *Hyalomma dromedarii*, also known as the poultry red mite [153]. The active molecules, notably (E)-caryophyllene and α-humulene, alone revealed a higher toxic effect than the whole essential oil and a lower toxic effect of myrcene. Although, it can be concluded that (E)-caryophyllene and α-humulene were responsible for the acaricidal effect of hemp essential oil, further investigation on the comprehensive understanding of the mechanisms of action underlying this effect as well as longitudinal studies with these active molecules needs to be established or documented as many active ingredients from *Cannabis*, which has shown to deter acari, have not been confidently ascertained.

In addition, this method continues to be advocated as a potential substitute for synthetic pesticides because of it being an environmentally friendly, low-cost, sustainable, and effective option in the fight against mites and ticks [153]. There are still very few rigorous studies undertaken concerning the acaricidal properties of *Cannabis* against ectoparasites of veterinary importance. Thus far, *R. microplus* remains the most studied tick across all the reviewed papers. 

Furthermore, even though roots and stem barks of *Cannabis* are ignored for their antimicrobial activities, they are characterized by common active compounds such as triterpenoids friedelin and epifriedelanol, which have been observed to be effective against a broad range of insect pest species on various plants such as Neem (*Azadirachta indica*), *Gynandropsis gynandra* and *Lavandula angustifolia* [102].

A laboratory experiment carried out on *A. indica* against *Tetranychus urticae* revealed the acaricidal properties attributed to *triterpenoids* [159]. Despite this empirical evidence from *A. indica,* no studies so far have evaluated the acaricidal properties of triterpenoids friedelin and epifriedelanol from *Cannabis*. Moreover, although the occurrence of the mites, such as the oribatid mite *Trhypochthoniellus longisetus*, has long been reported to colonize weak or stressed fishes, attaching to the mucous membranes and causing serious damage to farmed fish such as tilapia *Oreochromis niloticus*, it is very surprising that, to date, there are neither in vitro nor in vivo research studies documented which have investigated the acaricidal potential of *Cannabis* extracts and bioactive metabolites against fish mites [160]. This evidence proves that *Cannabis* remains one of many medicinal plants that are insufficiently tapped for their acclaimed pharmacological properties. 

### 3.10. Insecticidal Activities of Cannabis

Overall, various plants have shown potential to be used for their insecticidal and parasiticidal properties. Evidence suggests that essential oils and secondary metabolites of many plants have been largely exploited and proved to be effective against insect pests [161]. Among others, *Cannabis*-based insecticides and repellents were also traditionally employed in many parts of the world’s rural communities to protect crops from phytophagous arthropods [162]. Thus far, research studies focusing on the insecticidal activity of essential oils and active metabolites possessed by *Cannabis* have been undertaken as alternatives to effectively control ectoparasites of veterinary importance with great relevance, including *Ctenocephalides felis* [163]. In this respect, *Cannabis* essential oils and many other active metabolites have hitherto been under-researched, and it appears to have been limitedly exploited commercially.

Some previous papers report that *C. sativa* leaf essential oil demonstrated the presence of abundant terpenes and aliphatic compounds, which were observed to possess insecticidal activity in some medicinal plants such as *Azadirachta indica*, *Melia azedarach*, and *Lantana camara* [164]. Abé et al. [165] studied the effect of *C. sativa* essential oil against malaria vectors known as *Anopheles gambiae* and *Anopheles stephensi* and proved to be effective. Another scientific study documented that essential oil obtained from the fresh inflorescence of *C. sativa* displayed strong toxic effects against *Malus domestica* due to the presence of monoterpenes and sesquiterpenes [166]. These results proved to be eco-friendly as there was toxicity observed in non-targeted organisms [167]. However, the mechanism of action underlying this effect is still not clear nor elucidated. Although the insecticidal and antiparasitic activities of the essential oils and cannabinoid compounds have already been demonstrated in arthropods in the literature [168], it has not been demonstrated in some insects with medical importance, including bed bugs and water bugs. Other publications revealed that not only does the stickiness of *Cannabis* exudates trap insects, but it also possesses insecticidal and repellent activities [169]. Additionally, it may provide a possible synergistic mechanochemical defense in combination with the insecticidal phytocannabinoid acids, mainly tetrahydrocannabinolic acid (THCA), cannabidiolic acid (CBDA), and cannabichromenic acid (CBCA) [170]. Nevertheless, this synergistic mechanochemical with phytocannabinoids has not been evaluated against several insect pest species. Most literature reports on essential oils from inflorescences and leaves of *Cannabis* suggest that roots, stem barks, bud extracts, and their metabolites which were previously reported to possess antimicrobial properties are substantially untapped for their insecticidal activities. 

The essential oils from inflorescences have so far covered a large proportion of insect pests. For instance, Benelli et al. [153] further compared the activity of *Cannabis* essential oil from inflorescences against *Myzus persicae*, *Culex quinquefasciatus*, *Spodoptera littoralis,* and *M. domestica*. The results revealed high toxicity against *M. domestica* and *M. persicae*, moderate toxicity against the larvae of *S. littoralis,* and small toxicity against *C. quinquefasciatu*. Phytoconstituents analysis often reported to contribute to the insecticidal activity are major terpenes, such as β-myrcene and β-caryophyllene, obtained from inflorescences [171]. Most scientists assume that the higher prevalence of sesquiterpenes in the examined essential oils could be due to the drying process, which might have induced some chemical modifications in the composition of the starting material, including the evaporation of the low boiling-point compounds and occurrence of oxidative reactions, as in the conversion of β-caryophyllene in caryophyllene oxide, a major component of essential *Cannabis* oil [172]. Despite the literature showing positive activity against some insects, the ability of other essential oils obtained from the stem wood and the fresh and dry bark of *Cannabis* is often ignored in modern research. Based on this observation, it shows that exhaustive profiling of bioactive compounds and the chemical characterization and analysis of *Cannabis* compounds for their insecticidal activity has not been fully achieved. For instance, *Cannabis* has been observed to produce volatile sulfur compounds known to contribute significantly to the pungent aroma [173].

Although it is reported that this compound is produced in much lower concentrations than the major terpenes and sesquiterpenes [174], no study has broadly explored its insecticidal potential. This is even though some other plants such as garlic (*Allium sativum*), which is known to possess significant amounts of organosulfur compounds, have proven to contain strong insecticidal activity against *Aedes aegypti* and *Tribolium castaneum* [175]. Furthermore, research has proved that the effectiveness of plant-derived compounds, such as saponin, steroids, isoflavonoids, essential oils, alkaloids, and tannins, has the potential to exert insecticidal activities on various insects, notably mosquito larvicides [176]. 

Numerous investigations from various medicinal plants have shown the efficacy of these active compounds and essential oils as insect larvicides, repellents, over deterrents, and growth inhibitors without causing any harm to humans [177,178]. Surprisingly, details relating to their insecticidal properties on *Cannabis* have thus far not been sufficiently researched nor documented. Some previous reports suggest that the repellent connects with the female receptors, hindering the impression of smell and thus keeping the mosquitoes from perceiving the host [179].

In addition, researchers on mechanisms assume that these phytochemicals have the potential to change the physiology of insects in a variety of ways by affecting biochemical processes in insects, especially disrupting the endocrine balance [180]. Therefore, more in-depth in silico research studies focusing on the mechanisms should be carried out to produce concrete empirical evidence as most studies were conducted under in vitro conditions. Moreover, bioactive compounds are yet to be tested for their insecticidal activity. Their components must be included in silico analyses to know their absorption, distribution, metabolism, and excretion properties and possible human health risks [181]. According to Castillo-Morales et al. [182], who used computational analysis, the possible mechanism of action could be predicted for each essential oil or plant extract according to their components, which will significantly help to support its insecticide activity when in vivo experiments are validated. Table 1, Table 2 and Table 3 below summarize the different uses of *Cannabis* as anti-bacterial activity, antifungal, and anti-viral. 

## 4. Conclusions and Recommendations

*Cannabis* has proven to be a very potent plant known to have a diverse range of valuable secondary metabolites (phytochemicals). Therefore, its medicinal features, including untested molecules, should not be neglected nor overlooked. Many studies focused mostly on antibacterial properties, resulting in fewer studies analyzing antiviral, antifungal, anti-acaricidal, and anti-insecticidal properties. In addition, many studies analyzed the medicinal properties of *C*. *sativa*, thus neglecting possible medicinal properties that other *Cannabis* cultivars could possess. Furthermore, due to environmental issues associated with pesticide use and human health risks, many studies nowadays focus on finding alternatives to synthetic disease-control chemicals. *Cannabis* extracts and their phytochemicals have so far been demonstrated to be a solution. Therefore, more research studies investigating medicinal properties against diseases affecting livestock animals, poultry, and fish farming should take priority in a quest to discover alternative therapeutic agents instead of focusing on the same model organisms. In addition, the standard non-phytotoxic concentrations of these *Cannabis* extracts should be determined using MCSP to prevent any possible toxicity to the crops. Furthermore, *Cannabis* byproducts must be tested against the levels of cannabinoids, which will also contribute largely to the drug industry. Gas chromatography, mass spectra, and head space techniques should be taken into consideration as extraction methods and be included when determining and identifying volatile compounds possessed by the *Cannabis* plant. 

## Figures and Tables

**Figure 1 metabolites-14-00253-f001:**
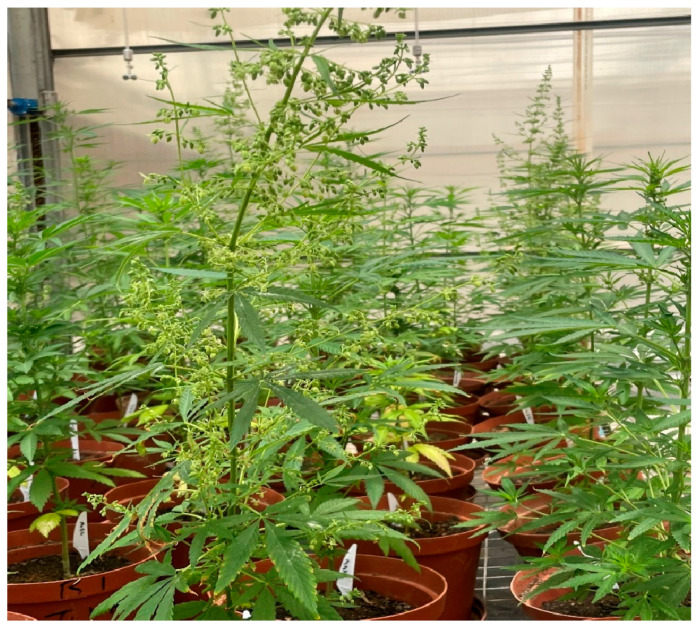
*Cannabis* plants at the flowering stage grown in the greenhouse (Nemadodzi L.E., February 2024).

**Figure 2 metabolites-14-00253-f002:**
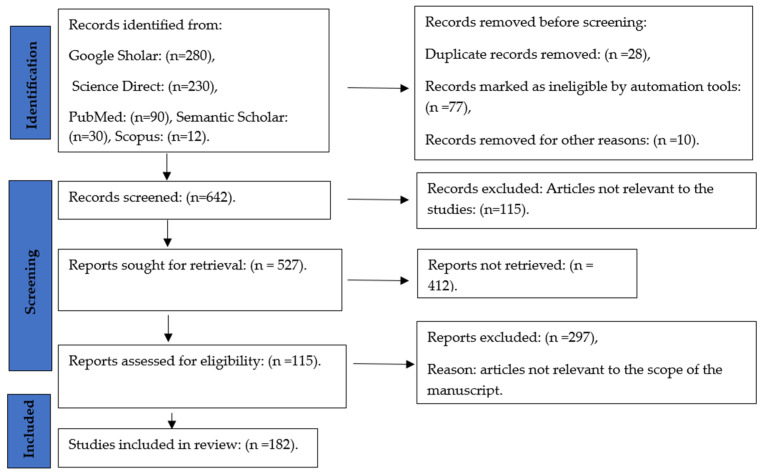
Identification of studies via databases and registers.

**Figure 3 metabolites-14-00253-f003:**
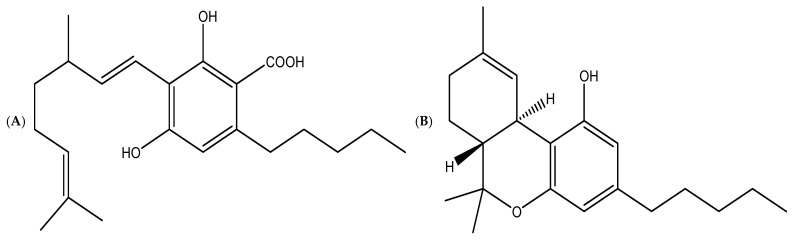
Chemical structure of the most common cannabinoids includes (**A**) Cannabigerolic acid (CBGA); (**B**) Delta-9-tetrahydrocannabinol (Δ^9^-THC); (**C**) Cannabidiol (CBD); (**D**) Cannabigerol (CBG); (**E**) Cannabinol (CBN); (**F**) Cannabichromene (CBC); (**G**) Cannabidiolic acid (CBDA); and (**H**) Tetrahydrocannabinolic acid A (THCA-A).

**Figure 4 metabolites-14-00253-f004:**
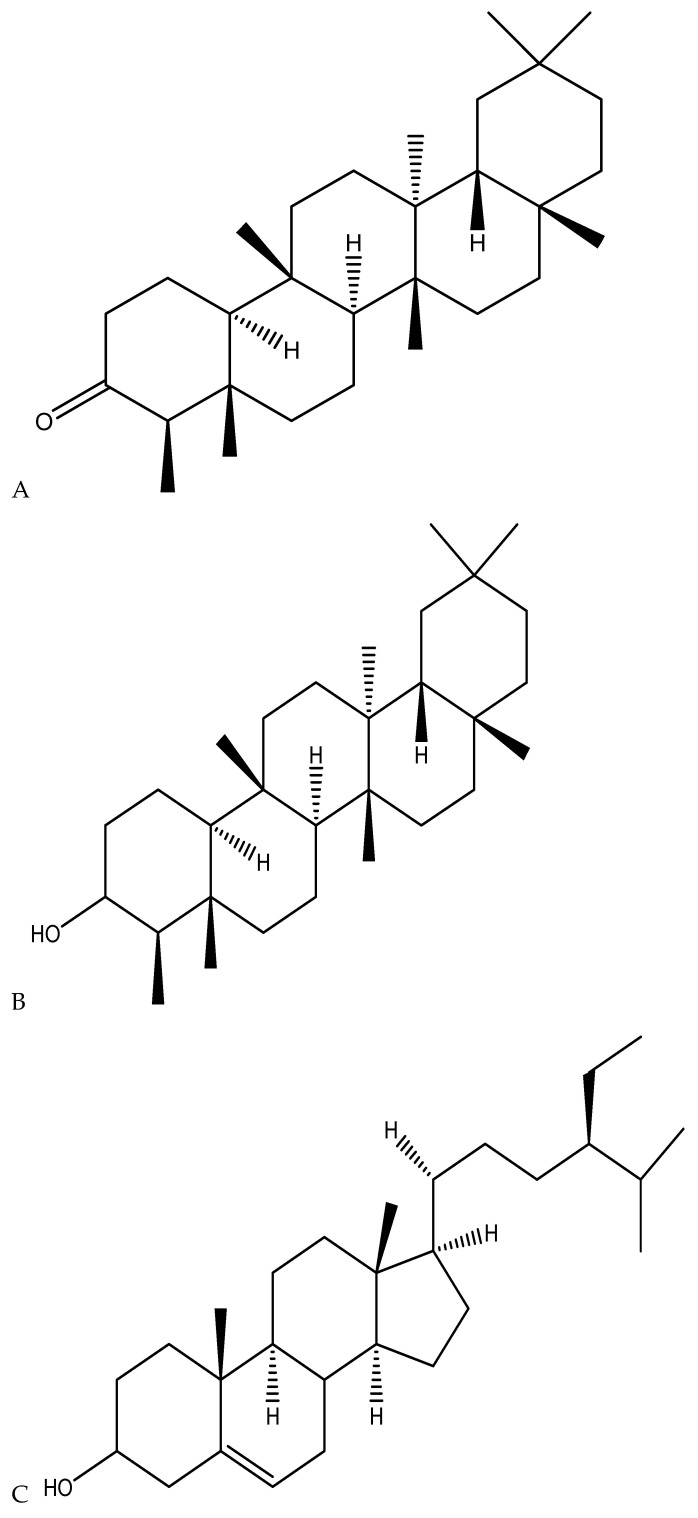
Chemical structure of (**A**) triterpenoids friedelin, (**B**) epifriedelanol, (**C**) sterols β-sitosterol, (**D**) stigmasterol, and (**E**) campesterol present in the *Cannabis* roots.

**Table 1 metabolites-14-00253-t001:** Summary of the antibacterial activity of *Cannabis*.

*Cannabis*	Target/Mechanism of Action	References
*Cannabis indica* (leaves, stems, and seeds extracts)	Possess strong anti-bacterial activity against multidrug-resistant bacterial strains such as *S. aureus*, *Bacillus cereus*, *E. coli* and *Klebsiella pneumoniae*, and *Pseudomonas aeruginosa*.	[16]
seed oil	Exerted strong antibacterial activity against *Bacillus subtili* and *S. aureus*.	[18]
CBV and THV	Exerts membrane disruption on bacterial species.	[111]
*Cannabis* leaf extract	Exerted the greatest antimicrobial effects on *S. aureus* 25923, with an inhibition zone of 14 mm.	[116]
Essential oils	Reduce the virulence of the food contaminant of *Listeria monocytogenes*.	[118]
CBD	Effective in reducing the bacterial colony count in dental plaque.	[120]
Crude alkaloid extracted from *Cannabis* leaf	Effectiveness against β strain of *E. coli* bacterial strains and the representative of skin, mouth, and ear microflora.	[123]

**Table 2 metabolites-14-00253-t002:** Summary of the antifungal and antinematicidal activity of *Cannabis*.

*Cannabis*	Target/Mechanism of Action	References
n-hexane fraction of *C. sativa*	Inhibit the growth and development of *Cryptococcus neoformans* known to be responsible for lung infection in humans with an IC50 value of 33.15 µg/mL.	[125]
*Cannabis* petroleum ether extract	Possess strong antifungal activity against *Candida albicans*.	[126]
n-butanol leaf extract of *C. sativa*	Inhibit the growth of *Aspergillus versicolor*	[127]
*Cannabis* inflorescence	Possess antifungal activity against *Alternaria* species	[131]
*C. sativa* leaf extracts	Significant reduction in number of galls, egg masses, nematode fecundity, and build.	[159]
Dried flowers and leaves of *Cannabis*	Effective in killing or repelling plant pathogenic nematodes.	[160]

**Table 3 metabolites-14-00253-t003:** Summary of the antiviral activity of *Cannabis*.

Cannabinoids/Compounds	Virus	Target/Mechanism of Action	References
CBD	HIV	Reduce extracellular vesicle release from HIV-infected monocytic cells and their viral cargo.	[137]
CBD	Hepatitis	Exert cytotoxicity effect on the liver cell line against hepatitis B and hepatitis C.	[138]
CBD	Kaposi sarcoma	Affect the proliferation and viability of Kaposi sarcoma through the VEGFR-3 signaling pathway	[140]
CBDA and CBGA	SARS-CoV-2	Interact with the SARS-CoV-2 spike protein S1 subunit and prevent the entry of several live viral variants into human epithelial cells	[144]
∆^9^-THC	SARS-CoV-2	Inhibits viral 3CLpro (IC50 3.62 μM)	[144]
CBV and ∆^9^-THCA-A	SARS-CoV-2	Molecular modeling predicts the inhibition of human ACE2	[149]
CBGA	SARS-CoV-2	Binds orthosterically and allosterically to the SARS-CoV-2 spike protein S1 subunit (Kd = 5.6 μM) and prevents cell entry of human epithelial cells and Vero cells by SARS-CoV-2 and early variants.	[144]

## Data Availability

No new data were created.

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
