# Peer review of "Evaluating the Metabolomic Profile and Anti-Pathogenic Properties of Cannabis Species"

_metabolites, 2024, doi:10.3390/metabo14050253_

Round 1

Reviewer 1 Report

Comments and Suggestions for Authors

The article reviews the chemical profile and antimicrobial properties of the Cannabis plant to benefit the health of people and vegetable crops. It is interesting reading and shows the need for more in-depth studies of the plant specie. To be accepted, authors must draw the chemical structures respecting the bond angles and improve the resolution of the figures in the text.

Author Response

Dear reviewer, 

The authors thank the reviewer for the positive comments and advice. As this manuscript is a review, we cannot make any changes/or amend the chemical structures, as they were taken from the original articles, and the authors acknowledged in the text (below the structures and in the reference list).

The figures used in the manuscript were taken with an advanced iPhone, we apologize if they are not as crystal clear.

We hope you will find our response in good favor.  

Reviewer 2 Report

Comments and Suggestions for Authors

The authors comprehensively reviewed metabolomics studies devoted to Cannabaceae plant species. An in-depth analysis of the main plant metabolites has been performed by the authors, using systematically researched reference data on the issue. The review-article is well-organized and the data are presented consistently. The contribution could be of interest in Metabolites' reader.

Comments on the Quality of English Language

Minor English editing is needed.

Author Response

Dear Editor, 

The authors would love to extend our gratitude to the positive comments. 

Reviewer 3 Report

Comments and Suggestions for Authors

Review “Evaluating the metabolomic profile and anti-microbial properties of Cannabis species” Monyela, S, Kayoka, P.N, Ngezimana, W, Nemadodzi, L.E. reviews various aspects of metabolomics of the genus Cannabis. It should immediately be noted that although the work is stated as a review, nevertheless, design elements of an experimental article are visible. Why are “Materials and Methods” needed here? I think this should be removed.

The works discussed are selected fresh. This is the strength of the review. The focus on practical applications of Cannabis metabolites also makes a good impression. The structural formulas of various Cannabis metabolites are quite appropriately given, which helps the inexperienced reader navigate this area. The authors consistently consider the impact of Cannabis compounds on various organisms harmful to humans (although viruses are difficult to consider as an organism), as well as on living organisms cultivated by humans, especially in agriculture. Here, authors need to pay attention to the title of their review, since “anti-microbial” is too narrow. After all, the review examines the impact on insects, nematodes, and mites, and these are clearly not microbes.

In addition, it would be desirable for the authors to add a section describing the main genes involved in cannabinoid metabolic pathways. Recently, data on these genes have already appeared. And mention the difference in the representation of cannabinoids in different representatives of the Cannabis genus.

An annoying drawback is Lines 199-211 and Lines 212-224 paragraph duplication.

In general, this work can be recommended for publication in Metabolites with minor revisions.

Author Response

Dear Reviewer, 

The authors thank the reviewer for the positive suggestions and comments. Below is a point-by-point response to the advice made. 

After thorough deliberation, the title of the manuscript was revised and now reads " Evaluating the metabolomics profile and anti-pathogenic properties of Cannabis species". 

Lines 199-211 emphasize the need to incorporate metabolomics in many plant science studies which will ultimately increase the number of metabolites whilst discovering unknown and yet unreported metabolites. This was followed by cannabinomics and analyzers which can be used. Also, the need to include environmental factors that may affect the growth of cannabis, cannabis edibles, drug discovery, and cosmetics is emphasized as compared to the usual research done on flowers and oil. The two paragraphs were included in the manuscript to create awareness and entice interest among researchers and scientists on the existing gaps in Cannabis that still need to be explored. 

The authors thank the reviewer for the great suggestion. The metabolomics pathway of cannabinoids is included in the revised manuscript under  section 3.4

The representations of cannabinoids are presented in section 3.4 and highlighted in yellow in the revised manuscript. 

We hope that our responses will find favour in you. 

Reviewer 4 Report

Comments and Suggestions for Authors

Dear Authors.

The article is very informative and presents a large volume of interesting scientific references; undoubtedly deserves to be published in such a respected scientific Journal.

Cannabis has proven to be a very potent plant known to have a diverse range of valuable secondary metabolites (phytochemical). Therefore, its medicinal features including untested molecules should not be neglected nor overlooked. Many studies focused mostly on antibacterial properties resulting in fewer studies analyzing antiviral, anti fungal, anti-insecticidal, and anti-acaridal properties. In addition, many studies analyzed the medicinal properties of C. sativa, thus neglecting possible medicinal properties that other Cannabis cultivars could possess. Furthermore, due to environmental issues associated with pesticide use and human health risks, many studies nowadays focus on finding alternatives to synthetic disease-control chemicals.

However, in order to polish the text of the article as best as possible, I would like to point out to the authors some inaccuracies and make possible corrections in the text of the article.

1. Line 400 Words: anti-insecticidal and anti-insecticidal . Are these words practically identical in meaning?

2.Unfortunately, the article does not reflect the fact that a large group of cannabinoids is also found in plants of other species. This fact must be noted.

3.It is also necessary to reflect in the article the most effective extraction methods for volatile cannabinoids today. Particularly noteworthy is the success of supercritical extraction in this area.

Author Response

Dear Reviewer, 

the authors thank the reviewer for the positive comments and advice. Below is the point-by-point response to the questions asked: 

the repetition of anti-insecticidal is deleted in the revised manuscript.

The requested information on the plant species that contain cannabinoids is included in the revised manuscript on track changes,  and highlighted in yellow under section 3.3. 

that authors thank the reviewers for the recommendation. However, as this is a review manuscript, the authors have included two extraction methods that can be used in the extraction of volatile compounds in today's world. 

We hope our responses will find favour in you.